# Molecular Dynamics Study on Behavior of Resist Molecules in UV-Nanoimprint Lithography Filling Process

**DOI:** 10.3390/nano12152554

**Published:** 2022-07-25

**Authors:** Jun Iwata, Tadashi Ando

**Affiliations:** Department of Applied Electronics, Faculty of Advanced Engineering, Tokyo University of Science, Tokyo 125-8585, Japan; 8121503@ed.tus.ac.jp

**Keywords:** nanoimprint lithography, UV-curable resin, molecular dynamics

## Abstract

In this study, we performed molecular dynamics (MD) simulations of the filling process of few-nanometer-wide trenches with various resist materials in ultraviolet nanoimprint lithography (UV-NIL) to identify the main molecular features necessary for a successful filling process. The 2- and 3-nm wide trenches were successfully filled with the resist materials that had (experimentally determined) viscosities less than 10 mPa·s. The resist composed of a three-armed bulky and highly viscous molecule could not fill the trenches. The radius of gyration of this molecule was smaller than half of the distance in which the first peak of its radial distribution function occurred. The available shapes of 1,6-hexanediol diacrylate (HDDA) and tri(propylene glycol) diacrylate (TPGDA), which are linear photopolymers, were compared to reveal that TPGDA is more flexible and adopts more conformations than HDDA. The terminal functional groups of TPGDA can be close due to its flexibility, which would increase the probability of intramolecular crosslinking of the molecule. This simulation result could explain the difference in hardness between the UV-cured HDDA and TPGDA based materials observed by experiments. The findings revealed by our MD simulations provide useful information for selecting and designing resists for fine patterning by UV-NIL.

## 1. Introduction

Nanoimprint lithography (NIL) is a promising, low-cost technology for fabricating nanoscale patterns in thin resists coated on substrates using compression molding, which is suitable for large-scale production [1,2]. Using NIL combined with high-resolution helium-ion beam patterning, Li et al. successfully transferred 4-nm wide half-pitch lines into a resist in 2012 [3]. Currently, NIL can be used to replicate the mold nanopattern regardless of the diffraction limit, leading to huge development in the fabrication of optical metasurfaces [4].

Understanding the underlying physical and chemical mechanisms in NIL processes is essential for further development of NIL. Theoretical and computational studies are powerful tools to gain new insights into cases not accessible by experiments. Hirai et al. used continuum-based simulation models to investigate a polymer deformation process in thermal NIL (T-NIL) [5]. Based on their simulation and experimental studies, they successfully demonstrated fabrication of a pattern with 100 nm width and 860 nm height using thick polymer by T-NIL [5]. They have also extensively studied ultraviolet NIL (UV-NIL) processes, such as resist filling and the mechanical properties and shrinkage of UV-cured resists based on conventional continuous mechanics [6]. Amirsadeghi conducted a numerical modeling study using the finite element method and reported that successful UV-NIL significantly depends on choosing an optimum concentration for the crosslinking agent [7]. These continuum-based simulation studies have provided deep insights into NIL with 100-nm resolutions. However, the pattern resolution of NIL is currently below 10 nm. The mechanical properties of these 10 nm-scale objects would be strongly affected by the molecular properties and behaviors of their resists. Therefore, atomic-scale analysis using molecular dynamics (MD) simulation of the NIL process would provide critical information for NIL processes with a sub-10 nm resolution. Taga et al. [8] and Yasuda et al. [9] performed MD simulations to investigate the relationships between the pressing force required to fill a mold cavity with a polymethylmethacrylate (PMMA) resist in a T-NIL process. This study showed a polymer-size effect on the NIL process, which did not appear in the continuum mechanics simulation. Kim et al. performed MD simulations to study the effect of PMMA polymer sizes on their filling behaviors of mold cavities with various geometries, showing that the radius of gyration (*R*_g_) of PMMA could be an indicator for selecting a suitable pattern size [10]. Several MD simulations of pressing nanoscale patterned stamps onto polymers were performed to study the molecular-level deformation behavior of resist polymers for T-NIL [11,12,13]. The adhesion between the mold and the PMMA polymer in the T-NIL process has also been simulated using MD methods [14,15,16]. In contrast to the polymers used in T-NIL, the behaviors of the short-chain resist molecules during the filling process in the UV-NIL are not well studied by MD simulations. Recently, we performed MD simulations of the filling process of four different resist materials used in UV-NIL, in which resists with viscosities lower than 10 mPa·s successfully filled 2- and 3-nm wide trenches, whereas those with viscosities higher than 92 mPa·s could not fill the trenches [17]. These atomistic simulation results well reproduced empirical rules obtained from experiments [17].

We performed MD simulations of the UV-NIL filling process with another resist with a viscosity of 10 mPa·s. Using the simulation data obtained in this study and our previous work, we further analyzed shapes, conformations, and distributions of UV curable resist molecules to investigate the effect of the molecular properties of the resist molecules on their UV-NIL processes.

## 2. Simulation Models and Methods

### 2.1. Resist Models

Four different resist materials, which consisted of variations of five molecules, were examined: N-vinyl-2-pyrrolidone (NVP); 1,6-Hexanediol diacrylate (HDDA); Tri(propylene glycol) diacrylate (TPGDA); Trimethylolpropane triacrylate (TMPTA); and 2,2-Dimethoxy-2-phenylacetophenone (DMPA). Table 1 lists the compositions and experimentally measured viscosities (*η*) of the four resists examined in this study. Figure 1 shows the structures of the molecules. NVP, HDDA, TPGDA, and TPMTA are UV curable monomers with vinyl or acryloyl groups, and DMPA is a polymerization initiator.

**Table 1 nanomaterials-12-02554-t001:** Compositions and measured viscosities (*η*) of the four resist materials examined in this study.

Resist	Composition ^1^	*η* [mPa·s]
Resist I	96% HDDA (1020), 4% DMPA (36)	4
Resist II ^2^	10% TMPTA (82), 57% TPGDA (448), 29% NVP (622), 4% DMPA (38)	8
Resist III	96% TMPTA (804), 4% DMPA (34)	95
Resist IV	96% TPGDA (816), 4% DMPA (38)	10

^1^ The number of molecules used in the simulation system for the resist filling process is shown in parenthesis. ^2^ From Ref. [18].

### 2.2. Force Field and Common Simulation Parameters

The general Amber force field (GAFF) [20] was adopted for the resist molecules with atomic charges determined by the Austin Model 1 with bond-charge correction [21,22]. The mold and substrate were made of a silicon (Si) diamond unit cell with a lattice constant of 5.43 Å. The bonds and bond angles of neighboring Si atoms were restrained by harmonic potentials with universal force field parameters [23]. The Lennard–Jones parameters of the Si atoms reported in Ref. [24] were adopted. For the van der Waals interactions, the Lorentz–Berthelot combination rules [25] were employed. All MD simulations used the periodic boundary conditions in all three directions. A cut-off length of 12 Å was used for van der Waals and short-range electrostatic interactions. The long-range electrostatic interactions were computed using the particle-mesh Ewald summation method [26]. A time step of 2 fs was used, and all bonds containing hydrogen atoms were constrained by the SHAKE algorithm [27]. A Langevin thermostat [28] or the Berendsen weak-coupling thermostat [29] was used for temperature control. For the Langevin thermostat, a collision frequency of 1.0 ps^−1^ was used. A relaxation time constant of 1.0 ps was used for the weak-coupling method. The Berendsen barostat [29] with a relaxation time of 1.0 ps was used for pressure control. Coordinates and energies of the systems were saved every 4 ps for post-analysis. All simulations were performed using Amber16 on GPU (GTX 1080Ti, NVIDIA, Santa Clara, CA, USA).

### 2.3. Estimation of Resist Contact Angles

Contact angles of the resists on the modeled Si surface were estimated using MD simulations of the resist cylindrical droplets on the surface to remove the line tension effects [30]. Figure 2a shows a schematic image of the system.

The simulations consisted of two steps. The first step is generating resist configurations in a bulk state, and the following is its procedure. (1) For each resist model, a total of ~600 resist molecules were randomly placed in a box of fixed size (*x* = 6.5 nm, *y* = 5.4 nm, and *z* = 13 nm) without significant overlap, where the number of molecules was calculated using the weight ratios listed in Table 1. The densities of the systems were roughly 0.5 g/L at this stage. (2) The system was minimized using the steepest descent method for the first 1000 steps followed by the conjugate gradient method for the subsequent 1000 steps. (3) *NPT*-ensemble MD simulations were performed at 298 K and 1 bar using Langevin thermostat and Berendsen barostat until the system was well equilibrated to converge the density to ~1 g/L. The pressure was controlled along the *z*-direction only, i.e., the dimensions in *x*- and *y*-directions were fixed.

In the second step, the resist generated in step one was placed on a surface of 26.1 × 5.4 × 11 nm^3^ Si substrate. The box size along the *z*-direction was 20 nm, which was three times greater than the height of the Si substrate and resist slab (~6 nm in total) so that slab–slab interactions were effectively turned off. Then, *NVT*-ensemble MD simulations were conducted at 298 K using the Langevin thermostat, where the positions of Si atoms at the bottom of the simulation box were fixed by a harmonic potential with a 100 kcal/mol/Å^2^ force constant. The simulation time depended on resist materials.

The density profile fitting procedure was used to calculate the contact angle [31]. The density profiles of resist molecules projected onto the *x*-*z* plane were computed using a grid size of 1 Å and averaged over 20 ns for resists I, II, and IV. For resist III, the density profiles were averaged over 50 ns, which was necessary for a stable computation. Then, a circular function was fitted on the interface contour data where the density data points with *z* < 12 Å were excluded. The last 100 ns simulation was divided into five blocks, each of length 20 ns, and the standard errors of the average were calculated to estimate statistical errors for resists I, II, and IV [32]. For resist III, the last 250 ns simulation was divided into five blocks, each of length 50 ns, for estimating a statistical error.

### 2.4. Estimation of Diffusion Coefficients

The self-diffusion coefficients, *D*, in the resists were estimated from the slope of the mean square displacement (MSD) of the molecules in the simulation boxes using the well-known Einstein relation:(1)$$6Dt=\underset{t\to \infty }{\lim}{(\left|r\left(t^{\prime }+t\right)-r\left(t\prime \right)\right|}^{2})$$
where **r**(*t*) is the molecule’s center of mass position at time *t*, and the brackets denote averaging over all molecules in the simulation box and time origins *t*’. Practically, *D* was estimated from the slope of the linear part of the MSD vs. time plot.

*NVT*-ensemble MD simulations of the bulk resist at 298 K were performed for 50 ns, which started from the coordinates generated as described in the first step in Section 2.3. The Berendsen weak-coupling thermostat [29] was used in the calculations for estimating *D* to eliminate effects of temperature control on transport properties [33]. Five independent MD simulations were performed for each resist, with different initial velocities to improve statistical accuracy and evaluate the statistical error. Then, *D* was evaluated from the slope of the MSD between *t* = 5 and 20 ns, where the initial part of the MSD line influenced by inertial effects was excluded.

### 2.5. Resist Filling Simulation

MD simulations of the resist filling process for resists I, II, and III were already reported in our previous work [17]. In this study, the simulations of resist IV were performed with the same simulation procedure as that reported in our previous work [17]. A schematic image of the resist filling simulation is shown in Figure 2b. The trench width, *∆*, was set to 2.17 and 3.26 nm. The simulation systems for the filling process were built by placing two resist boxes prepared at the first step in Section 2.3 along the *z*-direction between the mold and the substrate. The total number of atoms was about 48,000. Then, *NPT*-ensemble MD simulations at 298 K under 100 bar along the *z*-direction with fixed *x*- and *y*-dimensions were performed using the Langevin thermostat and Berendsen barostat. Simulation speed was ~70 ns/day. The shape characteristics of the resist molecules were analyzed during the last 15 ns (3750 frames) of the filling simulations. The properties of the resist molecules in bulk states were examined during the last 20 ns (5000 frames) of the MD simulations without mold/substrate as described at the first step in Section 2.3.

### 2.6. Radius of Gyration and Relative Shape Anisotropy

In order to check the structural characteristics of the conformations of the resist molecules, the 3 × 3 mass-weighted gyration tensor, ***T***, was calculated as
(2)$$T_{\alpha \beta }=\frac{1}{M}\overset{n}{\underset{i=0}{\sum }}m_{i}r_{i}^{\left(\alpha \right)}r_{i}^{\left(\beta \right)}\;\left(\alpha ,\beta =x,\;y,\;z\right),$$
where *M* is the mass of the molecule, *m_i_* is the mass of the *i*th atom, and *r_i_* is the coordinate of the *i*th atom relative to the center of the mass of the molecule. ***T*** is a symmetric positive definite matric, which can be diagonalized to obtain its three eigenvalues *λ*_1_ ≥ *λ*_2_ ≥ *λ*_3_. The squared radius of gyration $R_{g}^{2}$ is given by
(3)$$R_{g}^{2}=\lambda_{1}+\lambda_{2}+\lambda_{3}.$$The relative shape anisotropy, *κ*^2^, is defined by [34]
(4)$$\kappa^{2}=\frac{3}{2}\frac{\lambda_{1}^{2}+\lambda_{2}^{2}+\lambda_{3}^{2}}{\left(\lambda_{1}+\lambda_{2}+\lambda_{3}\right)}-\frac{1}{2}.$$

The *κ*^2^ ranges from 0 to 1 (one reflects an ideal linear chain and zero reflects symmetrical conformations, e.g., a sphere or a cube).

### 2.7. End-to-End Distance of Linear Chains

The end-to-end distances of linear resist molecules (HDDA and TPGDA) were measured using the distance between the terminal carbon atoms of the molecules.

## 3. Results and Discussion

### 3.1. Resist Contact Angles

In the NIL process, the wetting of the resist materials is an important property that should be considered. Using MD simulation, the contact angles of the resist droplets on the Si substrate were measured to evaluate the wetting properties of the model resists, as listed in Table 2. Appendix A shows snapshots of the systems at the end of the simulations for contact angle estimation, where the simulation length for each resist system was also listed. The contact angles of the four resists showed a trend of resist II < resist I = resist IV < resist III. The values were within the range of 104°–126°, and no significant difference was observed between them. The model resists have a weak tendency to dewet from the Si substrate.

### 3.2. Diffusion Coefficients of DMPA in Resists

The resist viscosity significantly impacts the filling rate. In general, low viscosity resists easily fill fine patterns. Computing viscosity by equilibrium MD simulations using the Green–Kubo formula is extremely difficult for higher viscosity systems [35]. Instead, we measured the self-diffusion coefficients of DMPA molecules in the four resists. The diffusion coefficient of a particle *D* is inversely proportional to the viscosity of surrounding fluid *η* via the Stokes–Einstein relationship, that is, D∝1/η for spherical objects in the dilute regime. Therefore, the values could be an index to represent the viscoelastic properties of the resist. Furthermore, all resists contain several DMPA molecules in the simulation systems, which helps increase statistical accuracy.

Table 2 lists the diffusion coefficients of DMAP *D* measured by MD simulations and the inverse ratios of *D* of resists to that of resist I. Table 2 also lists the ratio of the experimentally measured resist viscosities, *η,* to that of resist I for comparison. Appendix A shows MSDs of DMPA in the resists as a function of time. Importantly, the GAFF force field reproduced the order of diffusivities if the relation D∝1/η is valid for the examined four resists. For resist II, *D*_resist I_/*D*_resist II_ of 1.4 was close to the viscosity ratios, *η*_resist II_/*η*_resist I_, of 2.0. For resist III, *D*_resist I_/*D*_resist III_ of 36 was roughly close to *η*_resist III_/*η*_resist I_ of 24. For resist IV, however, *D*_resist I_/*D*_resist IV_ of 21 was eight times larger than the value *η*_resist IV_/*η*_resist I_, indicating GAFF overestimates the viscosity of resist IV. Therefore, the GAFF force field can reproduce the relative difference in viscosities of these resists except for resist IV.

### 3.3. Filling Behaviors

Figure 3 shows the changes in the volume of the resist IV simulation boxes as a function of the simulation time and snapshots of the systems at the end of the simulations. In ~8 μs, resist IV with a viscosity of 10 mPa·s filled the trench with *∆* of 3 nm (blue line in Figure 3a,b). In the case of the trench with *∆* of 2 nm, the system volume gradually decreased during the 8-μs simulation (red line in Figure 3a). The level of resist IV in the trench reached approximately two-thirds of the trench height (Figure 3c). Although the filling process of resist IV into the 2-nm-width trench was not completed within 8 μs, the resist would fill the trench within a finite time. The narrower trench widths required a longer time for the filling process. These results are consistent with the MD studies of PMMA, where the pressing force increased with the decrease in the mold cavity size [8].

MD simulations of the filling process for resists I, II, and III have already been reported in our previous paper [17]. Table 3 lists the times required for complete filling of the trenches with *∆* of 3 and 2 nm for resists I, II, and IV and their ratios to resist I. For resist IV with a viscosity of 95 mPa·s, the resist molecules neither fill the trenches with *∆* = 3 nor 2 nm [17]. The filling time of resist II into the trenches with *∆* = 3 and 2 nm were 2.3 and 3.8 times longer than resist I, respectively. These values are roughly comparable to the inverse ratio *D*_resist I_/*D*_resist II_ of 1.4. The filling time of resist IV into the trench with *∆* = 3 nm was approximately 52 times longer than that of resists I. This value is also roughly comparable to the inverse ratio *D*_resist I_/*D*_resist IV_ of 21. These results show that the resist viscosity largely determines the time required for filling few-nm-width trenches with resists in MD simulations. Measuring absolute viscosities of resists will be necessary to clarify the filling time–viscosity relation more clearly, which is our future work.

### 3.4. Shapes of the Resist Molecules in the Trenches and Bulk States

The effects of pressure and trench confinement on molecular shape were then investigated using *R*_g_ values of the resist molecules. Table 4 lists the *R*_g_ values of the resist molecules averaged over the same molecules and the last 15 ns of the filling simulations. Here, bulk *R*_g_ refers to the value calculated from the simulations without the mold and substrate under 1 bar. For comparison, Table 4 also lists the *R*_g_ values of the molecules with completely extended conformations shown in Figure 1. For resist II consisting of three UV-curing molecules, the 1-nm-wide trench was preferentially filled with the lower-viscosity NVP and TPGDA molecules as reported in our previous work [17]. It is worth mentioning that there was no molecular preference in filling the 3- and 2-nm-wide trenches (Appendix A). In the trenches with *∆* of 3 and 2 nm, the *R*_g_ values of resists I, II, and IV molecules are almost identical to those of the bulk states. The *R*_g_ values of TMPTA and TPGDA in resist II are also very close to those in resists III and IV, respectively. These results show that the molecular conformations of the resist materials are not affected by pressure, confinement, or mixing with the other resist molecules. HDDA and TPGDA are linear molecules. *R*_g_ of the HDDA in the resists decreased by 0.86 compared to the extended conformation. However, *R*_g_ of the TPGDA in the resists decreased by a factor of 0.74 compared to its extended conformation. Despite having a longer chain length than HDDA, TPGDA exhibited a more compact state than HDDA in most cases.

The molecular shapes of the resists were examined by measuring the relative shape anisotropy, *κ*^2^, of the multi-functional resist molecules. Figure 4 shows the distribution of *κ*^2^ of HDDA, TPGDA, and TMPTA in resists I, II, and IV. Figure 5 shows representative structures of these molecules in the MD simulations. According to Figure 4a,b, the distributions of *κ*^2^ of the linear HDDA molecule in the trenches with *∆* of 3 and 2 nm are close to those in the bulk state. The results indicate that the pressure and confinement in the trench do not affect the shape of the HDDA molecules. In the extended state (Figure 5a), the structures with *κ*^2^ = 0.8 are the dominant conformations. The distribution of *κ*^2^ in the TPGDA linear molecule used in resists II and IV evenly covers a *κ*^2^ range of 0.2–0.6 regardless of the presence of the trenches and the coexisting molecules (Figure 4c–f), which are different from the HDDA distribution. This result indicates that TPGDA can adopt various conformations from the compact to the extended states. HDDA and TPGDA have C6 alky and triproylene glycol chains, respectively, between the terminal acrylate groups. The difference in the chain flexibility between HDDA and TPGDA can be attributed to the fact that the energetic barrier of rotation around the C–O bond is lower than that around the C–C bond [36]. The TPGDA structure with *κ*^2^ = 0.2 is a compact conformation with two close terminal acryloyl groups (Figure 5b). The TPGDA structure with *κ*^2^ = 0.6 is a relatively extended conformation compared to that with *κ*^2^ = 0.2, and it has distant terminal acryloyl groups (Figure 5c). The distribution of the three-armed TMPTA molecule has two peaks at *κ*^2^ = 0.15 and 0.25 in bulk and the 3-nm-wide trench (Figure 4c,d). The two-peak phenomenon also occurred in resist III (Appendix A). In the 2-nm-wide trench, the peak at *κ*^2^ = 0.25 disappeared. The representative structures of the TMPTA with *κ*^2^ = 0.15 have a compact conformation compared to *κ*^2^ = 0.25 (Figure 5d,e). However, in contrast to TPGDA, the acryloyl groups in TMPTA are distant even in the compact conformation. This can be attributed to the very short branched chain connecting the acryloyl group, which almost resembles a rigid side chain. These results show that the bulky TMPTA molecules tend to adopt the more compact conformations in the narrower trench even though their conformational change is very small.

### 3.5. Radial Distribution Functions between Resist Molecules

The distribution properties of the resist molecules were analyzed by calculating the radial distribution functions, *g*(*r*), between the centers of masses of the molecules. As shown, similar molecular shapes (*R*_g_) and distributions were found both inside the trenches and in the bulk states. Thus, the *g*(*r*) values were calculated from the bulk state simulations to increase the statistical accuracy. The *g*(*r*) values between the centers of masses of the resist molecules in the four resist materials are shown in Figure 6. The *g*(*r*) values for TMPTA–TPGDA and TMPTA–TMPTA in resist II are statistically insufficient due to the small number of sampled pairs for these molecular pairs (Appendix A).

In resist I, the *g*(*r*) values of the HDDA molecules smoothly increase to match the bulk value and do not show clear peaks (Figure 6a), showing that the HDDA molecule does not occupy a specific coordinate in resist I. In resist II, the NVP–NVP and TMPTA–NVP pairs show a clear and intense first *g*(*r*) peak (~1.6) at a distance of 6 Å. However, the heights of the first *g*(*r*) peaks of the TPGDA–NVP and TPGDA–TPGDA pairs in resist II are less than 1.2 and broaden. This result is attributable to the heterogeneity of the molecular shape of TPGDA. The first *g*(*r*) peak of the TPGDA–NVP pair is located at a distance of 7.9 Å, which is 0.9 Å shorter than that of TPGDA–TPGDA (8.8 Å). These results indicate that the small NVP molecules surround the two-armed linear TPGDA molecules and the three-armed TMPTA molecules. In resist III, a clear first *g*(*r*) peak was observed at 6.9 Å (Figure 6c). In resist IV, TPGDA shows a broad first *g*(*r*) peak at ~10 Å with a height of only 1.1 (Figure 6d), indicating that resist IV is less structured than resists II and III.

The positions of the first *g*(*r*) peak and the sum of *R*_g_ values of all resist molecule pairs are listed in Table 5. In resists I, II, and IV, which successfully filled the 3-nm-wide trench and of which the viscosities are less than 10 mPa·s, the length of the first *g*(*r*) peak of the composing molecule pairs is higher than the sum of the *R*_g_ values of the pair. On the other hand, in the highly viscous material of resist III (95 mPa·s), which consists mainly of the three-armed TMPTA molecule, the length of the first *g*(*r*) peak of the TMPTA–TMPTA pair is smaller than the sum of the *R*_g_ values of the pair. Assuming that the molecules are spherical objects with a radius equivalent to *R*_g_, this result indicates that the TMPTA molecules overlap in the substrate. The overlap between the molecules would increase the viscosity of the material.

### 3.6. Radial Distribution Functions between the Functional Groups

The proximity of the acryloyl/vinyl groups in the resist materials is a major factor affecting the UV-curing process. Figure 7 shows the radial distribution functions (obtained from the bulk simulations) between the terminal carbon atoms of the acryloyl/vinyl groups in the four resist materials. This analysis only considered the intermolecular carbon atoms. The positions of the first peak of all resist materials on the graph are 4 Å. However, the resists exhibited different peak heights. Resist I (HDDA) and resist III (TMPTA) exhibited a height of 1.62. The height in resist II (NVP/TPGDA/TMPTA) is 1.57, slightly lower than in resists I and III. The height of the first peak (1.53) of resist IV (TPGDA) is lower than those of resists I and III. Figure 8 shows the probability distributions of the end-to-end distances of HDDA in resist I and TPGDA in resists II and IV. In HDDA, the probability of having an end-to-end distance of ~4 Å is negligible.

However, TPGDA can substantially adopt conformations with an end-to-end distance of ~4 Å because of its flexibility (Figure 4 and Figure 5), which lowers the height of the first *g*(*r*) peak between the terminal carbon atoms of the acryloyl/vinyl groups. During the UV-curing process in resists II and IV, which contain TPGDA, the probability of an intramolecular crosslinking would be higher than that in resists I and III. Hossain et al. reported that the hardness of thin films prepared using a urethane acrylate oligomer in combination with HDDA is approximately two times higher than that prepared using TPGDA, which was determined using the pendulum hardness test [37]. This difference in the hardness between the UV-cured thin films prepared using HDDA and TPGDA is attributable to the differences in the intramolecular crosslinking levels in their resists.

## 4. Conclusions

To reveal the molecular features required for a successful filling of few-nanometer-wide trenches with resists, all-atom MD simulations of a UV-NIL filling process of four resist materials composed of four photopolymers (i.e., HDDA, NPV, TPGDA, and TMPTA) and a photoinitiator (DMPA) were performed. The simulation results showed that resist materials containing HDDA, NVP/TPGDA/TMPTA, and TPGDA with viscosities less than 10 mPa·s successfully filled trenches with 2-nm and 3-nm widths. The three-armed bulky and highly viscous TMPTA molecule resist could not fill the trenches. Only the TMPTA–TMPTA pair had a first *g*(*r*) peak distance less than its sum of *R*_g_, according to the analysis of *R*_g_ and the radial distribution functions of the resist molecules. This metric could help in the selection and design of low viscosity photopolymers. Two linear photopolymers, HDDA and TPGDA, were compared. TPGDA is a more flexible molecule that is more likely to form compact states. As a result, the probability of having two close terminal acryloyl groups in the TPGDA molecule in the resist material is considerable, which would increase the probability of the intramolecular crosslinking in the UV-curing process, resulting in relatively lower hardness. The molecular features revealed in this computational study provide useful information for the future selection and design of resists in fine patterning using UV-NIL.

## Figures and Tables

**Figure 1 nanomaterials-12-02554-f001:**
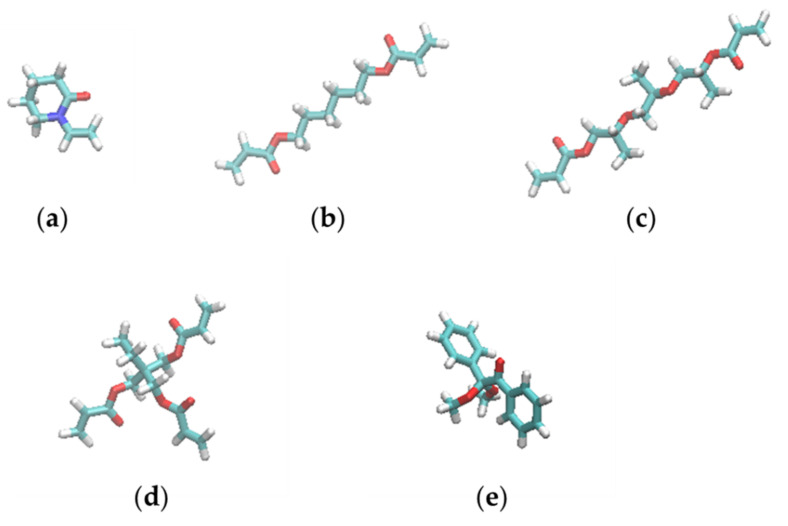
Structures of the molecules composing the resist materials used in this study: (**a**) NVP, (**b**) HDDA, (**c**) TPGDA, (**d**) TMPTA, and (**e**) DMPA. The figures were generated by visual molecular dynamics (VMD) [19].

**Figure 2 nanomaterials-12-02554-f002:**
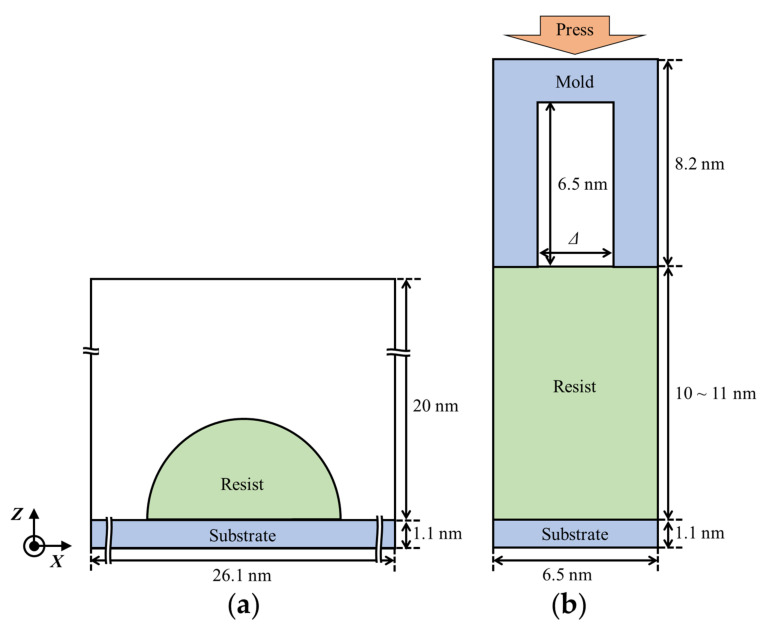
Schematics of the simulation systems for (**a**) contact angle estimation and (**b**) resist filling process. The box size along the *y*-direction was 5.4 nm for both systems. Mold and substrate were modeled with silicon atoms. The trench width *∆* in (**b**) was 2.17 or 3.26 nm.

**Figure 3 nanomaterials-12-02554-f003:**
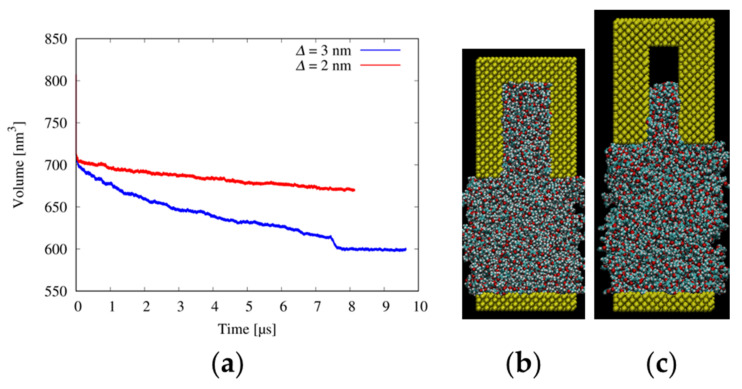
Filling simulation of resist IV. (**a**) Change in the system volume as a function of the simulation time, and snapshots of the systems at the end of the simulations with trench widths of (**b**) 3 and (**c**) 2 nm, respectively. In the snapshots, the molds are made from silicon atoms, and the substrates are illustrated using yellow spheres.

**Figure 4 nanomaterials-12-02554-f004:**
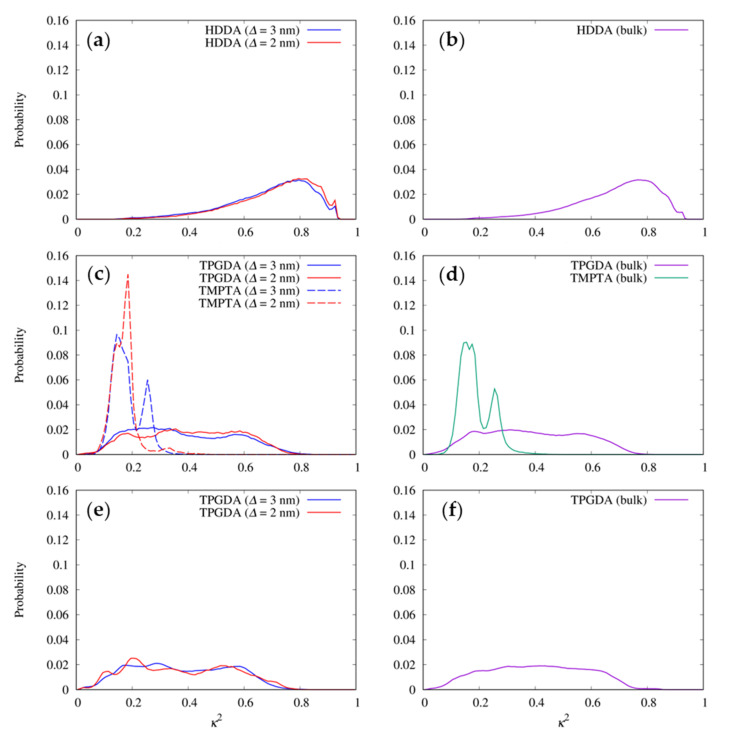
Distributions of the relative shape anisotropy, *κ*^2^, of the multi-functional resist molecules: HDDA of resist I in (**a**) the trenches and (**b**) bulk state; TPGDA and TMPTA of resist II in (**c**) the trenches and (**d**) bulk state; and TPGDA of resist IV in (**e**) the trenches and (**f**) bulk state.

**Figure 5 nanomaterials-12-02554-f005:**
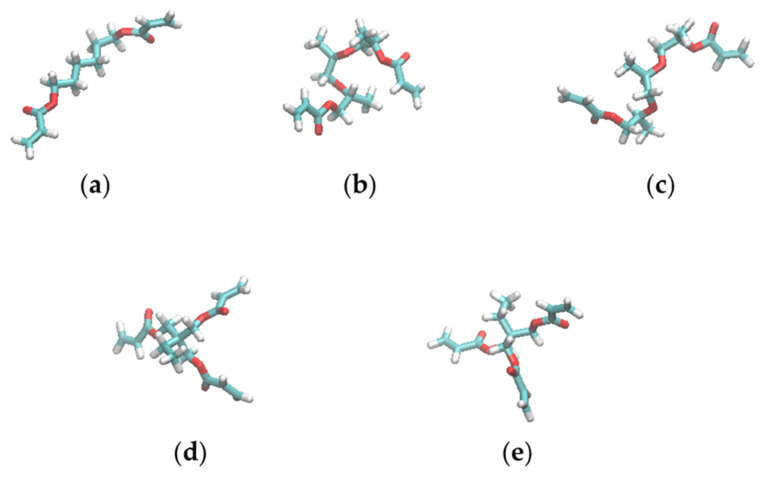
Representative structures of the multi-functional resist molecules in MD simulations. (**a**) HDDA with *κ*^2^ = 0.8, (**b**) TPGDA with *κ*^2^ = 0.2, (**c**) TPGDA with *κ*^2^ = 0.6, (**d**) TMPTA with *κ*^2^ = 0.15, and (**e**) TMPTA with *κ*^2^ = 0.25. The figures were generated by VMD [19].

**Figure 6 nanomaterials-12-02554-f006:**
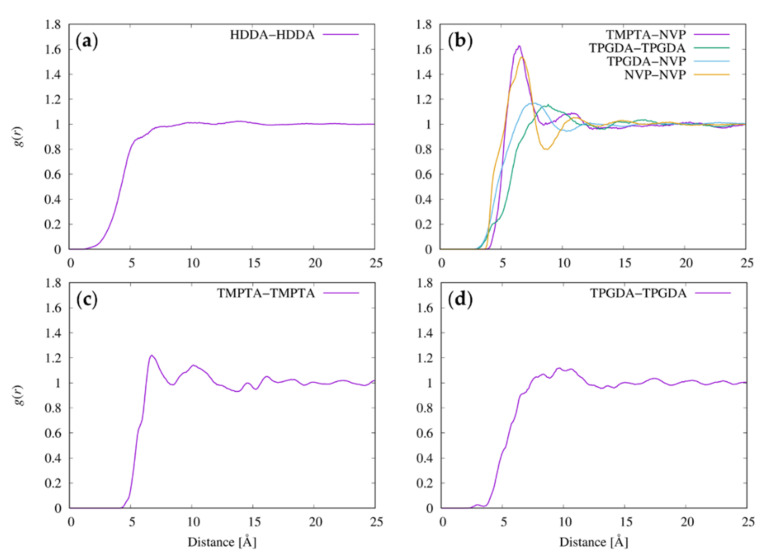
Pair radial distribution function, *g*(*r*), between the centers of masses of the resist molecules: (**a**) HDDA in resist I, (**b**) various molecule pairs in resist II, (**c**) TMPTA in resist III, and (**d**) TPGDA in resist IV.

**Figure 7 nanomaterials-12-02554-f007:**
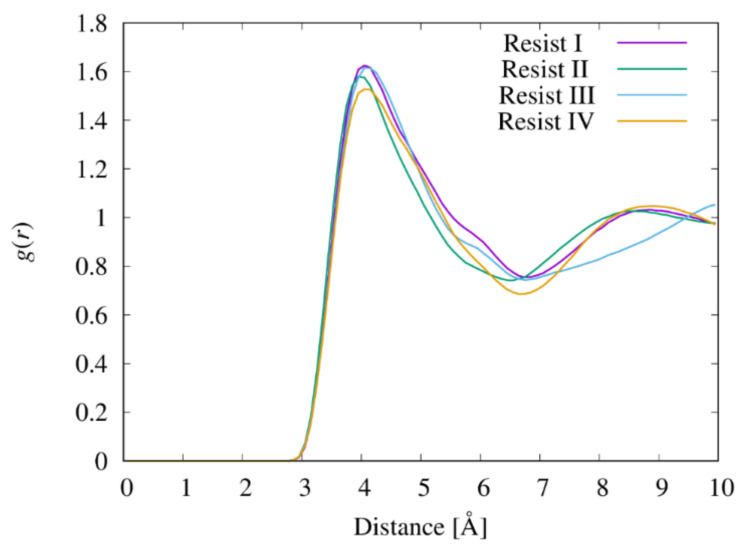
Radial distribution of the *g*(*r*) functions between the terminal carbon atoms of the acryloyl/vinyl groups in the four resist materials. Only the intermolecular pairs of the carbon atoms were considered in the calculations of *g*(*r*).

**Figure 8 nanomaterials-12-02554-f008:**
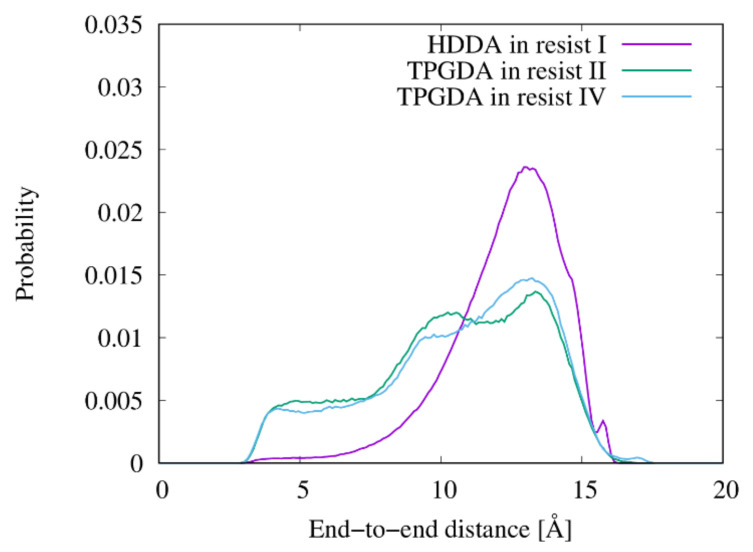
Probability distribution of the end-to-end distance of HDDA in resist I and TPGDA in resists II and IV.

**Table 2 nanomaterials-12-02554-t002:** Contact angles, diffusion coefficients of DMPA (*D*) estimated by MD simulations, the inverse ratio of *D* to resist I, and the ratio of the experimentally measured viscosity (*η*) to resist I.

Resist	Contact Angle [°]	*D* [10^−12^ m^2^/s]	*D*_resist I_/*D*	*η*/*η*_resist I_
Resist I	111 ± 1.8	4.79 ± 0.26	1.0	1.0
Resist II	104 ± 3.3	3.32 ± 0.31	1.4	2.0
Resist III	126 ± 2.1	0.13 ± 0.01	36	24
Resist IV	111 ± 0.3	0.22 ± 0.03	21	2.5

**Table 3 nanomaterials-12-02554-t003:** Time required for complete filling and its ratio to resist I for trench widths *∆* of 2 and 3 nm for various resists.

Resist	*∆* = 3 nm		*∆* = 2 nm	
	Time [μs]	Ratio to Resist I	Time [μs]	Ratio to Resist I
Resist I	0.15	1.0	0.25	1.0
Resist II	0.35	2.3	0.95	3.8
Resist IV	7.8	52	-	-

**Table 4 nanomaterials-12-02554-t004:** Radii of gyration of the resist molecules inside the trenches, bulk, and extended conformations.

Resist	Molecule	Radii of Gyration [Å]			
		*∆* = 3 nm ^1^	*∆* = 2 nm ^1^	Bulk ^1^	Extended Conformation ^2^
Resist I	HDDA	4.34	4.34	4.32	5.03
Resist II	TMPTA	3.76	3.75	3.75	3.67
	TPGDA	3.97	3.98	4.01	5.40
	NVP	2.07	2.07	2.07	2.01
Resist III	TMPTA	-	-	3.74	3.67
Resist IV	TPGDA	4.04	3.96	4.03	5.40

^1^ Values were averaged across the same molecules over the last 15 and 20 ns of the filling and bulk simulations, respectively. ^2^ Radii of gyration values were calculated using the structures illustrated in Figure 1.

**Table 5 nanomaterials-12-02554-t005:** Positions of the first peak of the radial distribution functions (*r**), the sum of radii of gyration (*σ_R_*_g_), and their differences, *δ* = *r** − *σ_R_*_g_, for various molecule pairs in the bulk states.

Resist	Pair of Molecules	*r** [Å]	*σ_R_*_g_ [Å]	*δ* [Å]
Resist I	HDDA–HDDA	- ^1^	8.6	-
Resist II	TMPTA–TMPTA	- ^1^	7.5	-
	TMPTA–TPGDA	8.4	7.8	0.6
	TMPTA–NVP	6.5	5.7	0.8
	TPGDA–TPGDA	8.8	8.0	0.8
	TPGDA–NVP	7.9	6.1	1.8
	NVP–NVP	6.7	4.1	2.6
Resist III	TMPTA–TMPTA	6.9	7.5	−0.6
Resist IV	TPGDA–TPGDA	9.6	8.0	1.6

^1^ No obvious peak.

## Data Availability

The data presented in this study are available from the corresponding authors upon reasonable request.

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
