# Peer review of "Molecular Dynamics Study on Behavior of Resist Molecules in UV-Nanoimprint Lithography Filling Process"

_nanomaterials, 2022, doi:10.3390/nano12152554_

Round 1
Reviewer 1 Report
The authors performed a computational study on modeling the filling process of four photopolymer blends (with different viscosities) with two different trench sizes. They analyzed the filling dynamics and the molecular composition/structure of inside/outside the trenches and found the external pressure and geometry of the trenches doesn’t affect the partitioning of the individual components during the filling process and in the equilibrium state. They analyzed the filling dynamics results and connect the molecular structures (conformations) of the photopolymer and provide some suggestions on the functional group of the photopolymer and the controllability of the filling dynamics. The authors want to provide some insights about how to control the filling dynamics by designing the chemical composition of the photopolymer but failed to provide a reliable logic reason to proof it. The simulation procedure looks OK, but the validation of the interaction parameters is missing which affect the reliability of the simulation results. The last but not the least, this manuscript is basically a continuation of their previous paper (ref 17, J Photopolym Sci Tec 2021, 34, 139-144.) with almost identical simulation target, setup and figures. But without providing more in-depth analysis and conclusion. Therefore, I don’t recommend this paper to be published on Nanomaterials in current form unless the authors address the following issues:
- The target simulation systems are built based on ref. 19, where the molecular composition is provided, and the viscosity data listed in table 1. The authors built the atomistic simulation using the composition from ref. 19 and using the GAFF to describe the interactions between atoms. However, the author didn’t check the viscosity of the simulation system. It is very important to check if the simulation system with current interaction parameters could match the experimental system or not. And viscosity is a good quantity that could be used as a target to describe the quality of the force field parameters and how reliable of your simulation data. Please adding the viscosity data obtained from simulation system used in this work and compare them with the experimental data.
- Line 115~116, the authors described the pressure is only applied to z direction and pressure is fixed along x and y direction. This is wrong, I guess the authors want to say the dimension in x and y direction is fixed.
- As a continuation of their previous work (ref 17, J Photopolym Sci Tec 2021, 34, 139-144), the authors shouldn’t copy the image and table directly from the published paper (yes, I saw the color of the schematic of the simulation has been changed, but the shape and text are quite similar). Besides the images, what make me complained most is the design and results are so similar between these two papers, which make me asking question if it is just one paper being splitting into two papers and published twice. These two papers study the same system with same setup and same method, the only difference is the trench sizes and some more analysis being added. Why not publishing them into one paper?
- In fig 3c, the snapshot of the simulation system with delta=2 nm looks weird, the picture shows some wrinkles on top, could the author check about it?
- In fig 3c, why the simulation stopped at 5 us for curve delta=2 nm. I think it is not in equilibrium state yet. Could the author add the missing simulation results?
- Regarding the driving force of the resist molecule filling the trench of the mold, the authors talked about the molecular structure such as gyration and end to end distance and try to explain the filling results using viscosity of those molecules. However, they didn’t talk about the most important factor which is the wetting/interfacial property of these molecules with the mold. For such a microscopic system, the contribution of surface energy or the wetting properties played a very important role in how filling happened and what is the driving force for this process. As a continuation of previous work, I hope the authors could go deeper to explore the real mechanism about the filling controllability instead of showing some results that are not quite important to understand this process.
- Language polishing is needed.
Reviewer 2 Report
This is an interesting work dealing with MD simulation of resist molecule in lithography filling process. Minor revisions are neeed to clarify the following questions:
(1) Why periodic boundary conditions were imposed in all directions, instead of, x- and y-direction only?
(2) Initial configurations were not mentioned. Were initial conditions randomized? Does initial configuration of resist molecules affect the simulation results?
(3) How many steps were taken in each trajectory for each resist? How many trajectories were taken for each set of simulations? What is the statistical error in the simulation result?
(4) Time evolution of system's volume change in microsecond period in Fig. 3 is the only dynamical information reported in this paper. Is that meant more than 109 simulation steps were taken for each resist filling process?
(5) Gyration ratio is a good index for comparison in most cases for nonlinear shape resist. But nor for HDDA. Is there any other index for rod-shape resist?
Round 2
Reviewer 1 Report
The markers in the revised pdf is super messy which make it difficult for me to read line by line, but I do checked the revised part and all points are addressed. Language polishing is needed. Publish as is.
